# The AFF-1 exoplasmic fusogen is required for endocytic scission and seamless tube elongation

Fabien Soulavie[1], David H. Hall [2] & Meera V. Sundaram[1]

Many membranes must merge during cellular trafficking, but fusion and fission events initiating at exoplasmic (non-cytosolic) membrane surfaces are not well understood. Here we show that the *C. elegans* cell–cell fusogen anchor-cell fusion failure 1 (AFF-1) is required for membrane trafficking events during development of a seamless unicellular tube. EGF-Ras-ERK signaling upregulates AFF-1 expression in the excretory duct tube to promote tube auto-fusion and subsequent lumen elongation. AFF-1 is required for scission of basal endocytic compartments and for apically directed exocytosis to extend the apical membrane. Lumen elongation also requires the transcytosis factor Rab11, but occurs independently of dynamin and clathrin. These results support a transcytosis model of seamless tube lumen growth and show that cell–cell fusogens also can play roles in intracellular membrane trafficking events.

[1] Department of Genetics, University of Pennsylvania Perelman School of Medicine, Philadelphia, PA 19104, USA. [2] Department of Neuroscience, Albert Einstein College of Medicine, Bronx, NY 10461, USA. Correspondence and requests for materials should be addressed to M.V.S. (email: sundaram@pennmedicine.upenn.edu)

Some biological tubes, including many capillaries in the vertebrate vascular system, are so narrow that the lumen is contained within a single-cell body[1–3] (Fig. 1). Such unicellular tubes can be seamed (sealed with an autocellular junction) or seamless (lacking junction along the tube length). In some cases, seamless tubes can grow enormously to adopt quite complex, branched or elongated shapes, as exemplified by terminal tracheal cells in *Drosophila*, or the excretory canal cell in *Caenorhabditis elegans*[1]. Despite its unusual topology, a seamless tube shares fundamental properties with larger tubes, including an internal apical (luminal) domain, an outer basal domain, and cell–cell junctions that link the tube to its neighbors in the organ. Key questions are how the intracellular apical domain of a seamless tube is generated, how it grows, and how it is appropriately shaped and stabilized. Loss of narrow capillaries is associated with cardiovascular and neurological syndromes such as small vessel or microvascular disease, hereditary hemorrhagic telangiecstasia and cerebral cavernous malformation[4–6], so understanding how seamless tubes develop has considerable health relevance.

Both endocytic and exocytic mechanisms have been proposed to generate and shape the intracellular lumens of seamless tubes, but the origin of the apical membrane and the specific trafficking pathways involved remain poorly understood[1]. Studies of mammalian endothelial cells cultured in vitro suggested that intracellular lumens form through macropinocytosis ("cell gulping")[7], a specific type of membrane ruffling-associated, clathrin-independent, endocytosis[8]. Macropinocytosis generates large internal vacuoles that appear to subsequently merge within and between cells to form a continuous lumen[7,9]. On the other hand, studies of seamless tubes in the zebrafish vascular system, *Drosophila* trachea and *C. elegans* excretory system have supported models involving polarized exocytic trafficking[10–14]. In these models, cell–cell contacts and/or membrane invagination nucleate a new apical domain at one edge of the cell, which then expands and grows inward based on exocytic vesicle-dependent delivery of apical membrane components[1]. However, the precise identity and origin of these exocytic vesicles remains unclear. Finally, other studies in *C. elegans* revealed that a seamless tube can form by cell wrapping and self-contact to form a seamed tube with an autocellular junction, followed by auto-fusion to eliminate the junction and convert to a seamless toroid[2,15–17]. Auto-fusion may be a widely used mechanism, since it also generates some seamless tubes in the zebrafish vascular system[18] and in mammalian epithelial cells grown on micropillar arrays[19]; however, relevant fusogens have not yet been identified in vertebrates. The endocytic, exocytic, and auto-fusion-dependent models of seamless tube formation are not mutually exclusive, and all three mechanisms could be involved in generating the elongated lumens and complex shapes of many seamless tubes in vivo.

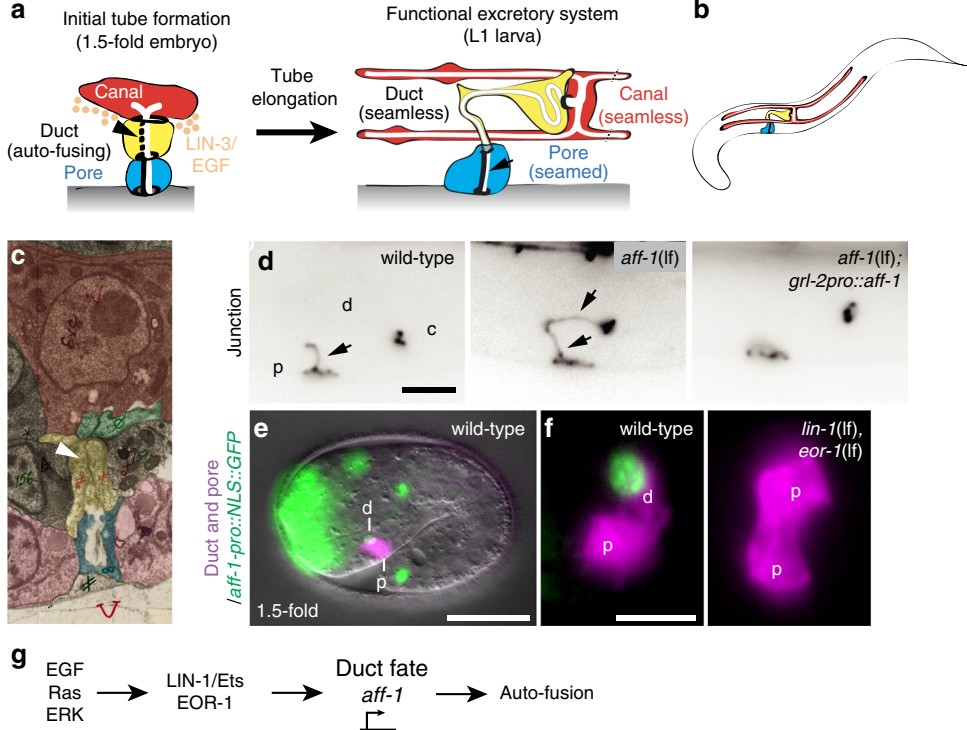

**Fig. 1** EGF-Ras signaling drives *aff-1* expression to promote auto-fusion in the excretory duct. **a** Schematics of excretory system tube development. Prior to the 1.5-fold stage of embryogenesis, the excretory duct (yellow) and pore (blue) cells each wrap to form unicellular tubes by establishing self-contacts and auto-junctions. A LIN-3/EGF signal is emanating from the excretory canal cell (in red). The excretory pore cell maintains its auto-junction (thick dark line) and the excretory duct auto-fuses (arrowhead and dotted line), forming a seamless tube. After tube formation the cell morphologies change, with elongation of the duct and canal cells. **b** Schematic of an early L1 larva showing the three tube cells of the excretory system. **c** Transmission electron micrograph of a 1.5-fold stage embryo showing excretory cells, from unpublished data by C. Norris and D. H. Hall. Prior to auto-fusion, the duct in yellow has an auto-junction (white arrowhead). In red, the canal cell, in green an excretory gland cell, in blue, the pore cell and in pink hypodermal cells. **d** Excretory system junctions visualized with *ajm-1::gfp*. In wild-type, the pore (p) has an auto-junction (arrow), but the duct (d) and canal (c) do not. In *aff-1* (tm2214) mutants, both duct and pore cells have auto-junctions (arrows). In *aff-1*(tm2214) mutants with the transgene *grl-2pro::aff-1* expressed in duct and pore, both cells auto-fuse. **e** Wild-type 1.5-fold embryo-expressing *aff-1pro*::NLS-GFP (green) in the duct (d) but not pore (p). **f** Unlike wild-type, in *lin-1* (n304) *eor-1*(cs28) double mutants, both cells adopt a pore-like morphology (ref. [30] and Supplementary Fig. 2) and neither cell expresses *aff-1pro*::NLS-GFP. **g** EGF-Ras-ERK signaling promotes duct fate and *aff-1* expression. Scale bar, **d**, **f** = 5 μm and **e** = 10 μm

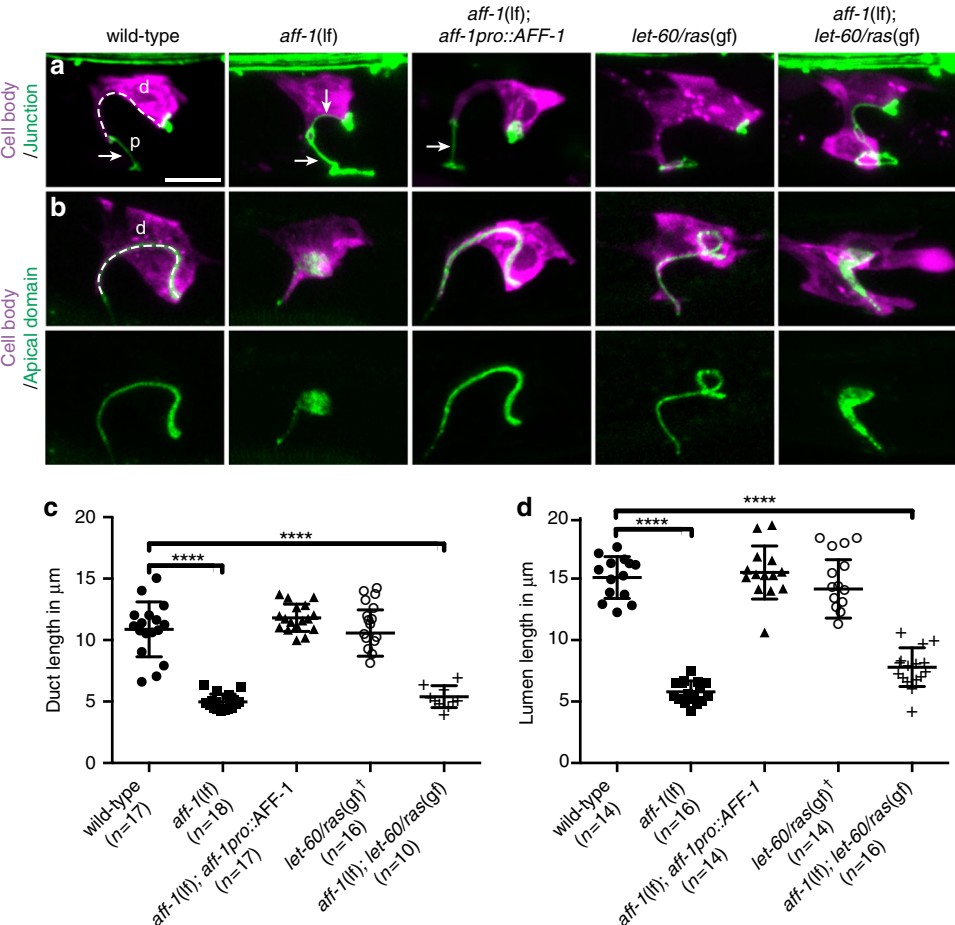

**Fig. 2** AFF-1 is necessary for duct cell and lumen elongation. **a**, **b** Confocal Z-projections of L1 stage larvae: d, duct; p, pore. Duct cell is labeled with *lin-48pro::mRFP* (magenta). Cell junctions are labeled with *ajm-1::GFP* (green, **a**), or apical domains are labeled with *let-653::SfGFP* (green, **b**). Arrows indicate autocellular junction. The excretory duct cell body and lumen have an elongated shape in wild-type, *aff-1* rescue and *let-60(n1046gf)*, but are shorter in *aff-1(tm2214)* and *aff-1(tm2214)*; *let-60(n1046gf)*. **c** Measurements of duct cell length tracing from the pore-duct junction to the duct–canal junction, as demonstrated with the dotted line in "**a**" wild-type. Error bars = ± standard deviation (SD). **d** Measurements of duct lumen length, tracing as indicated with the dotted line in "**b**" wild-type. † = measure on a binucleate cell. Scale bar = 5 μm. **** = *p*-value < 0.0001, Mann–Whitney test

In *C. elegans*, seamless tube auto-fusion is mediated by homotypic interactions between the exoplasmic fusogens epithelial fusion failure 1 (EFF-1) or anchor-cell fusion failure 1 (AFF-1)[15,16], single-pass transmembrane proteins that also mediate many cell–cell fusion events[20–23]. EFF-1 and AFF-1 belong to a widely conserved structural family that also includes viral class II fusogens[24,25] and the HAP2/GCS1 gamete fusogens of plants and protists[26–29]. Here, we describe new roles for AFF-1 in endocytic scission and apically directed exocytosis for intracellular lumen elongation. Our results support a transcytosis model of seamless tube lumen growth and show that cell–cell fusogens also can play roles in intracellular membrane trafficking events.

## Results

### EGF-Ras-ERK signaling promotes excretory duct cell auto-fusion and shaping.

Receptor tyrosine kinase signaling through Ras and ERK promotes development and shaping of many seamless tubes, including the *C. elegans* excretory duct tube[30]. The duct is the middle tube of three tandemly connected unicellular tubes in the excretory system, a simple osmoregulatory organ[31] (Fig. 1a, b). During excretory system development, LIN-3/EGF expressed by the excretory canal cell acts through Ras-ERK signaling and two nuclear targets, LIN-1 (an Ets factor)[32,33]

and EOR-1 (a BTB-zinc finger protein)[34,35], to promote excretory duct (seamless tube) vs. pore (seamed tube) cell identity[30] (Fig. 1g and Supplementary Fig. 1). Both tube types initially have simple shapes and autocellular junctions, but only the duct auto-fuses to eliminate its junction and become seamless[16] (Fig. 1a and Supplementary Fig. 1). Transmission electron microscopy (TEM) and confocal imaging of junctions indicated that duct auto-fusion occurs at around the 1.5-fold stage of embryogenesis, within an hour after tube formation (Fig. 1c and Supplementary Fig. 1). Subsequently, the duct tube elongates and adopts an asymmetric shape, with a long, narrow process that connects it to the pore tube (Figs. 1a and 2). The duct lumen becomes longer than the cell itself, taking a winding path through the cell body (Figs. 1a and 2). Ras signaling is both necessary and sufficient for duct vs. pore fate, auto-fusion and shaping[30] (Supplementary Fig. 1), but how the intracellular lumen elongates remains poorly understood.

### EGF-Ras-ERK signaling upregulates *aff-1* expression to stimulate duct auto-fusion.

Duct auto-fusion requires the fusogen AFF-1[16] (Fig. 1d), leading us to hypothesize that Ras signaling may promote AFF-1 expression or activity. A transcriptional reporter that fuses 5.4 kb of the *aff-1* upstream genomic sequence (*aff-1pro*) to nuclear-localized green fluorescent protein (NLS-GFP) was expressed in the duct beginning at

the 1.5-fold stage of embryogenesis, around the time when auto-fusion occurs, but was never observed in the pore (Fig. 1e, f). Duct expression of *aff-1pro::NLS-GFP* required the Ras guanine nucleotide exchange factor SOS-1 and redundant contributions of the nuclear factors LIN-1 and EOR-1 (Fig. 1f and Supplementary Fig. 2). When *aff-1pro* was used to drive expression of an *aff-1* cDNA, it rescued the auto-fusion defects of *aff-1* mutants (Supplementary Fig. 2). Ectopic expression of *aff-1* in both the duct and pore, using the *grl-2* promoter, was sufficient to induce pore auto-fusion and pore-duct fusion in wild-type (WT), *aff-1* (loss of function (lf)), *and sos-1* (thermo-sensitive (ts)) mutant backgrounds (Fig. 1d and Supplementary Fig. 2). Altogether, these data indicate that Ras signaling upregulates *aff-1* expression to drive duct auto-fusion (Fig. 1g).

**AFF-1 also is required for duct tube elongation and apically directed trafficking**. We found that subsequent duct tube elongation also requires AFF-1. In *aff-1* mutants, the duct cell has a very short process, and the lumen is only a third of its normal length (Fig. 2). Both phenotypes can be rescued by *aff-1pro::AFF-1* (Fig. 2). The *aff-1* short duct phenotype is epistatic to *let-60 ras* (gf) (Fig. 2), consistent with AFF-1 acting downstream of Ras signaling. Furthermore, *aff-1* mutants accumulate apical markers in an expanded domain adjacent to the lumen (Fig. 2b). Confocal and super-resolution stimulated emission depletion (STED) microscopy revealed that this domain corresponds to numerous distinct puncta (Fig. 3a–c), suggesting accumulation of vesicular trafficking intermediates. Similar patterns were observed with three different markers, the luminal matrix protein LET-653[36], the apical tetraspan protein RDY-2, and the vacuolar ATPase subunit VHA-5[37], suggesting broad dysregulation of apically directed trafficking in *aff-1* mutants.

To test if AFF-1 is sufficient to promote tube elongation, we examined animals carrying the *grl-2pro::AFF-1* transgene described above. Otherwise WT animals-expressing *grl-2pro::AFF-1* had a binucleate tube with a duct-like shape and a long lumen (Supplementary Fig. 3), similar to *let-60/ras*(gain of function (gf)) mutants (Fig. 2a). However, *sos-1* (ts) mutants-expressing *grl-2pro::AFF-1* had a binucleate tube with a lumen only slightly longer than in *sos-1*(ts) single mutants (Supplementary Fig. 3). Therefore, *aff-1* is just one of multiple Ras targets required for duct tube elongation and shaping.

**AFF-1 promotes lumen elongation independently of its role in auto-junction removal**. *aff-1* mutant apical trafficking defects could be a secondary consequence of auto-fusion failure, as previously proposed for *eff-1* mutants[38], or could reflect a direct role for AFF-1 in membrane trafficking events. To distinguish between these possibilities, we used the ZIF-1-dependent proteolysis system[39] to remove AFF-1 protein after auto-fusion was complete (Fig. 4 and Supplementary Fig. 4). The ZF1 degron was engineered into the endogenous *aff-1* locus using CRISPR-Cas9-mediated genome editing[40], and the ZIF-1 protease was expressed in the duct at different developmental stages using transgenes with different promoters. Positive control experiments confirmed that AFF-1::ZF1 was functional, and that early AFF-1 degradation (using *grl-2pro::ZIF-1*) abolished duct auto-fusion, reduced lumen length, and expanded apical domain width (Supplementary Fig. 4). Later AFF-1::ZF1 degradation (using the heat-shock promoter *hsp-16.41pro::ZIF-1*) did not affect auto-fusion, but still reproduced the apical domain phenotypes observed in *aff-1*(lf), including reduced lumen length and expanded apical domain width (Fig. 4). We conclude that AFF-1 plays a direct role in apically directed trafficking that is temporally separable from its role in auto-fusion.

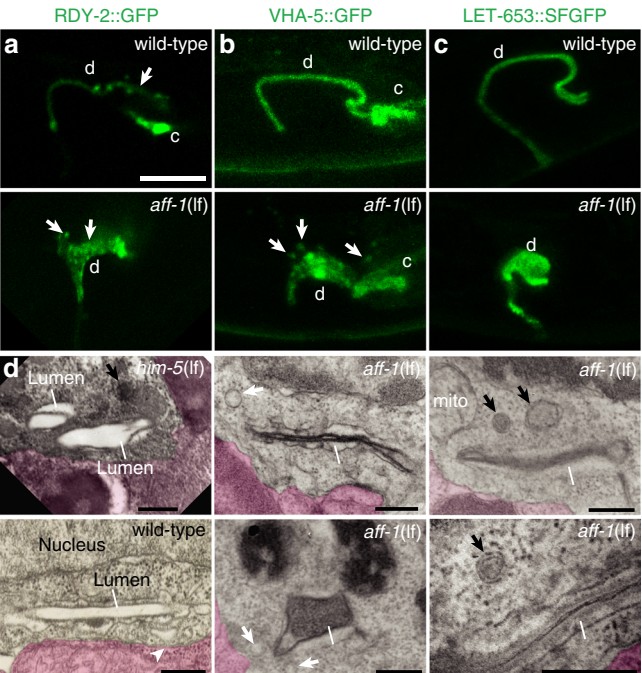

**Fig. 3** *aff-1* mutants accumulate apically marked vesicles. **a** Super-resolution stimulated emission depletion (STED) microscopy slices and **b**, **c** confocal Z-projections of L1 stage larvae: d, duct; c, canal. Apical markers are **a** tetraspan protein RDY-2[37], **b** vacuolar ATPase subunit VHA-5[37], and **c** luminal matrix protein LET-653[36]. In wild-type, apical signal is highly restricted to a region near the elongated lumen. *aff-1*(tm2214) mutants show a shorter and wider apical domain, with isolated puncta as shown by arrows. **d** TEM transverse slices of normal [*him-5*(e1490) or N2] or *aff-1* (tm2214) L1 duct. Neighboring cells are false-colored in pink. Line indicates cuticle-lined lumen. Arrowhead indicates possible endocytic cup in wild-type. Small spherical vesicles (white arrows) and larger multi-membrane objects (arrows) are located near the lumen in *aff-1* mutants. Scale bars, **a**–**c** = 5 µm; **d** = 300 nm

**aff-1 mutant duct cells exhibit a block in basal endocytic scission**. Next, we examined both apical and basal membranes and overall ultrastructure of *aff-1*(lf) mutant duct cells by TEM of serial sections. In four L1 specimens examined, the duct lumen was similar in diameter to wild-type (155 nm ± 30 (n = 4) in *aff-1* (lf) vs. 170 nm ± 40 (n = 4) in WT, Fig. 3d), although some regions were filled by abnormal darkly staining material in addition to the normal cuticle lining (Fig. 3d). Small vesicles and more complex lysosome- or autophagosome-like objects were present near the lumen (Fig. 3d), some of which likely correspond to the abnormal apical compartments observed by confocal microscopy (Fig. 3a–c). Most dramatically, the duct cell body contained large inclusions, similar in size to the nucleus, that consisted of highly convoluted, narrow (~30 nm) membrane tubules (Fig. 5a). Analysis of serial sections suggested that these inclusions were continuous with the basal plasma membrane (Fig. 5a and Supplementary Fig. 5). Similar membrane inclusions were also observed in some epidermal cells of *aff-1* mutants (Supplementary Fig. 5), but were never observed in WT specimens (n = 4).

The *aff-1* basal inclusions resemble a blocked endocytic intermediate. To further assess this possibility, we exposed WT and *aff-1* mutants to FM4-64, a membrane-binding styryl dye that can enter cells only via endocytosis[41,42]. After 30 min of exposure, WT L1 animals had little or no dye inside the duct or pore cell bodies, but after 150 min of exposure, much more dye had entered the interior of both cells, consistent with active

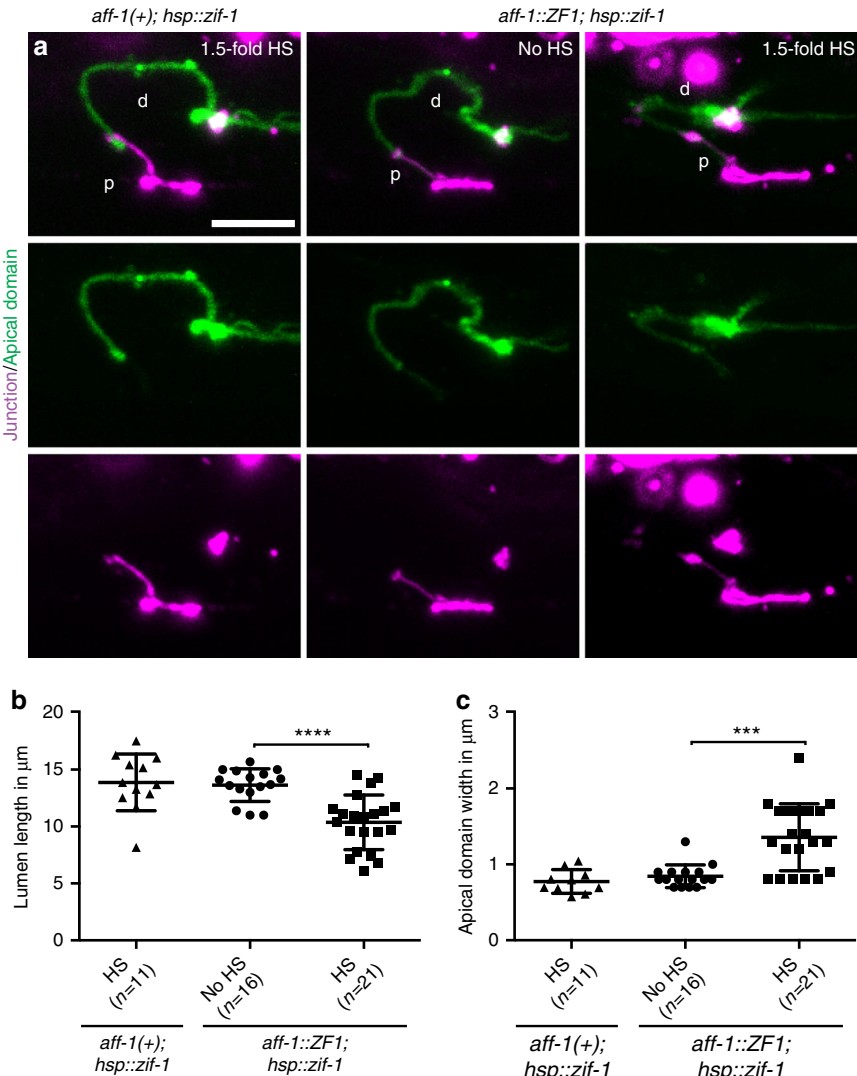

**Fig. 4** AFF-1 promotes lumen elongation independently of junction removal. **a** Confocal Z-projections of L1 stage larvae-expressing apical marker *rdy-2* (*cs233*, [*rdy-2::gfp*]) in green and junction marker *ajm-1::mCherry* in magenta. d, duct; p, pore. Control animals *aff-1*( + ) or without heat shock (HS) have a normal lumen morphology. *hsp*::ZIF-1-induced degradation of AFF-1::ZF1 after heat shock (HS) at the 1.5-fold stage led to a shorter and wider apical domain. **b**, **c** Measurements of lumen length and apical domain width, respectively, in animals with normal duct auto-fusion. Error bars = ± SD. Scale bar = 5 μm. **** = *p*-value < 0.0001, *** = *p*-value < 0.001, Mann–Whitney test

endocytosis (Supplementary Fig. 6). In duct/pore-specific *aff-1::ZF1* mutants after just 10 min of exposure, the dye-marked internal regions of the duct (Fig. 5b). These results were confirmed by additional observations at the L4 stage (Supplementary Fig. 6). Furthermore, fluorescence recovery after photobleaching (FRAP) experiments indicated that the dye-marked compartments in *aff-1* duct cells recovered rapidly from photobleaching (Fig. 5d and Supplementary Fig. 6). Altogether, the TEM, FM4-64, and FRAP experiments suggest that *aff-1* mutant duct cells have extensive internal membrane compartments that are connected to the basal plasma membrane (Fig. 5e), consistent with a defect in endocytic scission.

**AFF-1 localizes to sites of auto-fusion and basal endocytosis**. If AFF-1 directly mediates endocytic scission, then it should localize to the neck of internalizing vesicles at the basal plasma membrane. To visualize AFF-1 protein, we examined transgenic animals expressing an AFF-1::mCherry fusion under control of the 5.4 kb *aff-1* promoter described above. AFF-1::mCherry is not fusion competent, so its pattern of localization must be

interpreted with caution, but we note that fusion-incompetent versions of the paralog EFF-1 accumulate more robustly than functional versions at sites of membrane fusion[43]. In 1.5–2-fold embryos, around the time of auto-fusion, AFF-1::mCherry localized specifically to duct apical membranes (Fig. 6a). In later embryos and larvae, AFF-1::mCherry relocated and accumulated in puncta throughout the duct cell, most of which were located at or near the basal plasma membrane by L1 stage (Fig. 6a, b). To test if the basal puncta correspond to sites of endocytosis, we repeated the FM4-64 dye experiments in the AFF-1::mCherry strain. Under imaging conditions where internalizing FM4-64-positive vesicles could be observed in WT larvae, 37/59 of such vesicles (*n* = 19 larvae) were accompanied by a basal spot of AFF-1::mCherry (Fig. 6d, e). We conclude that AFF-1 is appropriately positioned to mediate endocytic scission.

**Duct lumen elongation is dynamin- and clathrin-independent but requires the recycling endosome protein RAB-11**. The previous results demonstrate that AFF-1 is required for endocytic

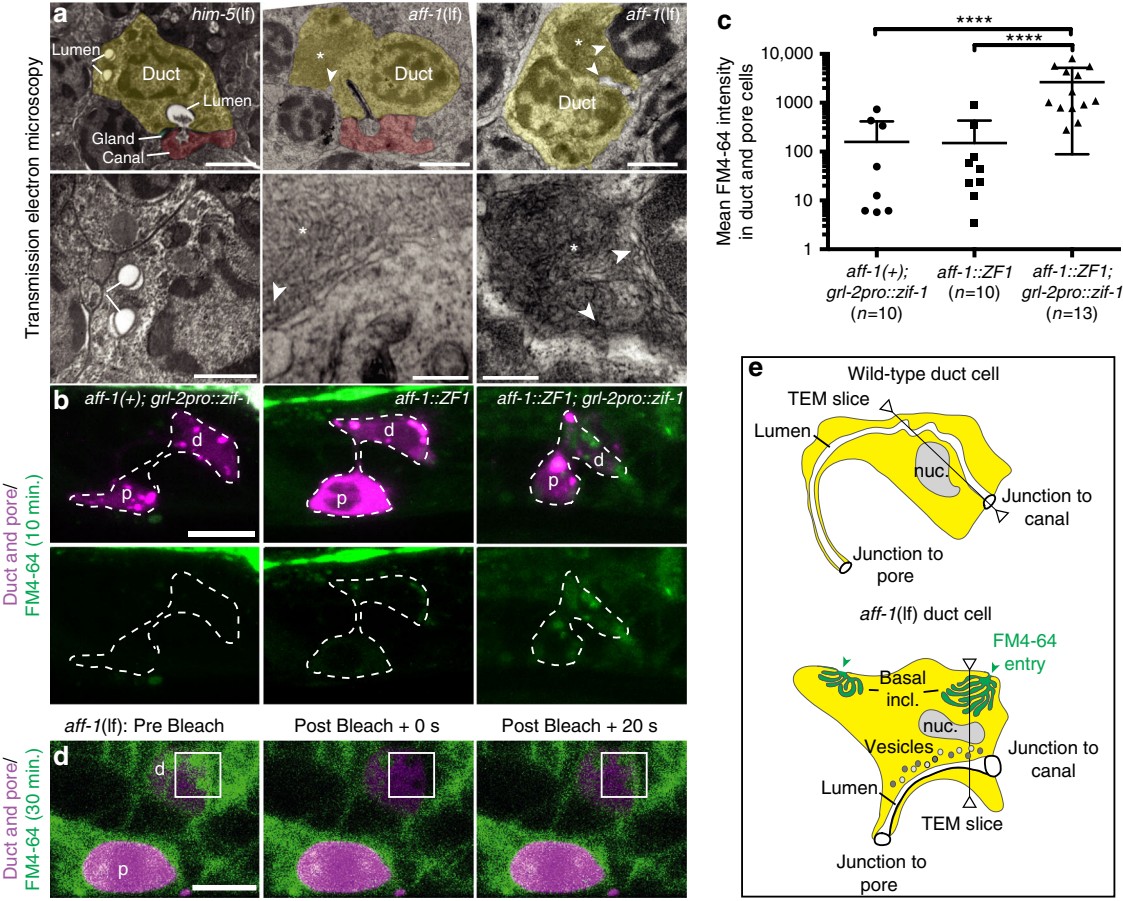

**Fig. 5** AFF-1 is required for endocytic scission. **a** Transmission electron micrographs, showing transversal sections of the excretory duct cell body in L1 larvae. Duct is post-colored in yellow and canal cell is post-colored in red when applicable. Dorsal side up. Lower panels show a higher magnification of upper panels. The control (*him-5(e1490)*) duct has a lumen crossing the section three times and a continuous basal membrane. Both *aff-1(tm2214)* mutant duct cell bodies have one or no lumen crossings but contain large membrane inclusions (asterisks). White arrowheads indicate regions where the inclusion appears continuous with the basal plasma membrane; these regions are magnified in lower panels. See also Supplementary Fig. 5. **b** FM4-64 dye filling assays in L1 animals. Confocal slices are shown, with duct (d) and pore (p) cells in magenta and FM4-64 in green. Left panels: *grl-2pro::zif-1::mCherry* control. Middle panels: *aff-1::zf1* control, with duct and pore cells marked by *grl-2pro::mCherry*. Right panels: *aff-1::zf1; grl-2pro::zif-1::mCherry* for duct/pore-specific degradation of AFF-1. AFF-1-depleted cells show a very strong FM4-64 signal compared to the controls. **c** Quantification of dye fluorescence intensity, plotted on a logarithmic scale. **** = *p*-value < 0.0001, Mann–Whitney test. Error bars = ± SD. **d** Fluorescence recovery after photobleaching (FRAP) experiment on *aff-1(tm2214)* mutant duct cell at L1 stage. FM4-64-marked inclusions within the bleached region of interest (white square) recovered rapidly. See Supplementary Fig. 6 for further details. **e** Schematic interpretation. Scale bars = 1 μm (**a** upper panels), 300 nm (**a** lower panels) and 5 μm (**b**, **d**)

vesicle scission and for apically directed membrane trafficking to promote duct lumen elongation.

To understand which specific trafficking pathways are involved in duct lumen elongation, we observed lumen length in various endocytosis and cell trafficking mutants. Duct lumen elongation occurred normally in temperature-sensitive mutants for *dyn-1*/dynamin and *chc-1*/clathrin, as well as in null mutants for the early endosome component RAB-5 (Fig. 7a, b), suggesting that lumen elongation occurs independently of clathrin-mediated endocytosis. However, *rab-5* mutants had a disorganized and widened apical domain (Fig. 7a, c), consistent with a role for RAB-5 in constraining lumen width, as has been reported for seamless tubes in *Drosophila*[44]. The most dramatic effect on duct lumen length was seen in mutants for RAB-11, a key player in endosome recycling and transcytosis[45,46] (Fig. 7a, b). These results suggest that duct lumen elongation requires a transcytosis mechanism to add membrane to the intracellular apical domain (Fig. 7d).

## Discussion

Fusogens of the class II structural family include EFF-1 and AFF-1 in *C. elegans*[24], HAP2/GCS1 in many lower eukaryotes and plants[27–29], and the fusion proteins of certain enveloped viruses such as Zika, dengue, yellow fever, and West Nile[25,47]. Given their wide phylogenetic distribution and poor sequence-level conservation, it is possible that additional, unrecognized members of this family exist in vertebrates. These single-pass transmembrane proteins mediate cell–cell fusion events to form syncytial tissues[20–22], fuse gametes[26], and allow viral infection of host cells[25]. EFF-1 and AFF-1 can also mediate cell auto-fusion to shape or repair neuronal dendrites and axons and to generate narrow seamless tubes with intracellular lumens[2,15,16,48–52].

Our results reveal a new and unexpected requirement for *C. elegans* AFF-1 in membrane trafficking events important for intracellular lumen growth. In addition to retaining inappropriate autocellular junctions in a tube that should be seamless, *aff-1* mutants fail to elongate this tube, show broad dysregulation of apically directed trafficking, and accumulate extensive internal

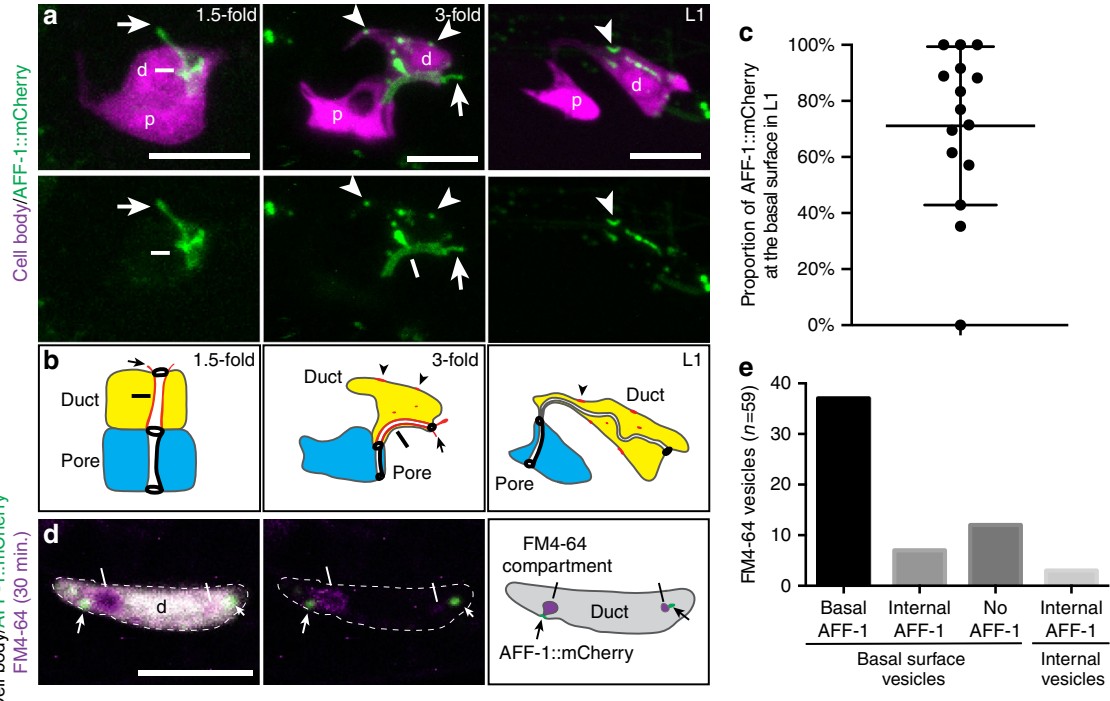

**Fig. 6** AFF-1 localizes to sites of auto-fusion and basal endocytosis. **a** Confocal Z-projections at different developmental stages in wild-type, d, duct; p, pore. The excretory duct and pore cell bodies are labeled with *grl-2pro::YFP* (magenta) and AFF-1 localization visualized with *aff-1pro::aff-1::mCherry* (green). At the time of duct auto-fusion, in 1.5-fold stage animals, AFF-1::mCherry localizes predominantly at the apical surface of the duct cell (line). The signal also extends dorsally (arrow); since the duct is the only *aff-1* expressing cell in this region at this stage (Fig. 1e), the extension presumably corresponds to an extension of the duct apical domain into a neighboring cell such as the excretory canal tube or excretory gland, with which the duct lumen connects[31]. The localization of AFF-1::mCherry progressively shifts to become cytoplasmic and basal (arrowheads) in later stages. In L1 stage, AFF-1::mCherry is still present >6 h after duct auto-fusion. **b** Schematic interpretation. **c** Volocity quantification of the proportion of AFF-1::mCherry at the basal membrane in L1 larvae. Error bars = ± SD. **d** Confocal single slice of a wild-type L1 larva. AFF-1::mCherry (green) localizes adjacent to FM4-64-marked endocytosing vesicles (magenta and white bar) at the basal membrane of the duct cell (gray). **e** Quantification of the four categories of FM4-64 positive vesicles. Scale bar = 5 μm

membranes continuous with the basal plasma membrane. The requirement for AFF-1 in membrane trafficking is genetically and temporally separable from the requirement in junction removal, and during lumen elongation, AFF-1 fusions accumulate at sites of basal endocytosis. We propose that AFF-1 directly mediates endocytic scission during transcytosis-mediated seamless tube lumen growth.

Membranes must merge during many biological processes, including cellular trafficking. In some cases, such as vesicle fusion, contact between merging membranes initiates at the cytosolic (endoplasmic) side; soluble N-ethylmaleimide-sensitive factor (NSF) attachment protein (SNAP) receptors (SNAREs) and other endoplasmic membrane fusogens have been extensively studied, and are required to overcome repulsive hydrostatic forces to bring adjacent vesicle membranes closer than 10 nm for fusion[23,53]. In other cases, such as cell–cell fusion, membrane merging initiates at the non-cytosolic (exoplasmic) side; here, exoplasmic fusogens such as HAP2 are needed to bring adjacent cells' plasma membranes closer than 10 nm for fusion[23,26]. Although endocytic scission involves fission rather than fusion, it is another example of a membrane merging event that initiates at exoplasmic membrane surfaces[2,54]. However, the mechanisms underlying scission are not well understood, and are generally thought to involve forces applied from the endoplasmic side of the membrane[55,56]. For example, the small GTPase dynamin promotes scission of clathrin-coated vesicles[8], and the BAR-domain protein endophilin promotes scission of some uncoated tubulovesicle

compartments[57]. Our results suggest that, in at least some cases, cell–cell fusogens can mediate scission during clathrin-independent endocytosis.

A direct role for AFF-1 in endocytic scission is consistent with its known activity as a cell–cell fusogen that is both necessary and sufficient to merge membrane bilayers[21,58]. Furthermore, cell–cell fusogens are appropriately oriented in cellular membranes, with their fusogenic domains extending into non-cytosolic spaces such as extracellular environments[24,59] (Fig. 7d). Cell–cell fusogens require other forces to bring membranes into close proximity, but once two membranes are within ~10 nm, the fusogens can engage to merge them[23]. We propose that forces that drive membrane invagination and tubulation during endocytosis could be sufficient to allow AFF-1 fusogen engagement when AFF-1 is present on the plasma membrane (Fig. 7d). In this way, AFF-1 would cooperate with other cytoskeletal or membrane-bending machineries to drive the final stages of membrane scission.

Cell–cell fusion and endocytic scission could be mechanistically linked in some cases, since cell–cell fusion removes plasma membrane, which may need to be endocytosed and recycled[60]. Indeed, vesicles have been observed near some (though not all) fusing plasma membranes in *C. elegans*[38,61,62]. Several fusogen mutants, including *C. elegans eff-1* and *Tetrahymena hap2*, have previously been found to accumulate abnormal vesicles near unfused plasma membranes, but these vesicles were proposed to be secondary consequences of fusion failure[38,63]. We found that abnormal vesicles in *aff-1* mutants accumulate independently of auto-fusion failure, and, therefore, reflect a more direct

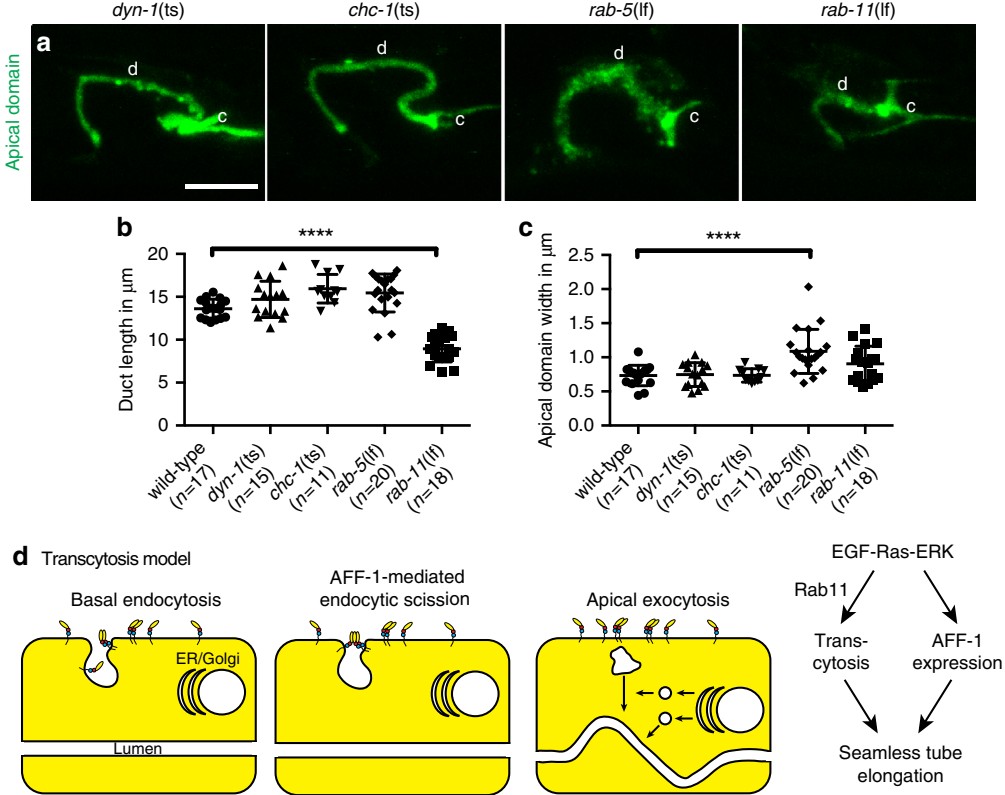

**Fig. 7** Duct lumen elongation requires RAB-11. **a** Confocal Z-projections of L1 larvae-expressing apical domain marker RDY-2::GFP. d, duct; c, canal. *dyn-1(ky51ts)* and *chc-1(b1025ts)* mutants resemble wild-type (see Fig. 2b). In *rab-5(ok2605)* mutants, the apical signal is disorganized and widened. In *rab-11(tm2063)* mutants, the apical domain is shorter and compared to wild-type. **b**, **c** Measurements of duct length and apical domain width, respectively. **** = *p*-value < 0.0001. Mann–Whitney test. Scale bar = 5 μm. Error bars = ± SD. **d** Transcytosis model, showing endocytic scission events mediated by AFF-1

requirement in membrane trafficking. Furthermore, we provided evidence that AFF-1 is required for scission of endocytic vesicles at a basal plasma membrane surface that does not participate in cell–cell fusion events. Similarly, Ghose et al.[64] have independently shown that the fusogen EFF-1 promotes a specific phagosome sealing event. Therefore, cell–cell fusogens can be re-purposed for endocytic scission events that occur in the absence of cell–cell fusion.

We propose a transcytosis model for duct tube growth that combines all three previously proposed mechanisms for seamless tube formation, with nucleation of an initial lumen by wrapping and auto-fusion, and then growth of the lumen by endocytosis from the basal surface, followed by exocytosis to the apical surface (Fig. 7d). This model is consistent with the observed Rab11 requirement, the presence of both endocytic and exocytic blocks in *aff-1* mutants, and with observations that EGF signaling can stimulate apically directed transcytosis in mammalian epithelial cells[45]. According to this model, EGF signaling turns on AFF-1 expression to promote duct tube auto-fusion, and also stimulates a clathrin-independent form of endocytosis at the duct tube basal membrane. AFF-1 mediates vesicle scission to resolve the endocytic compartments into discrete internal vesicles, which then undergo Rab11-dependent transcytosis to add to the apical membrane. The basal inclusions observed in *aff-1* mutants would then reflect continued rounds of endocytosis despite a failure to detach endocytosed membranes from the basal plasma membrane. We hypothesize that the exocytic block in *aff-1* mutants is an indirect consequence of the endocytic block—for example, Golgi-derived vesicles may accumulate aberrantly due to absence of appropriate partner vesicles for exocytosis. Alternatively, AFF-1 could play a direct role in some exocytic scission mechanism, but further studies will be needed to address that possibility.

More than 15 years ago, Podbilewicz[60] proposed a "fusomorphogenic hypothesis" in which one role of developmental cell fusion is to redistribute membrane from basolateral to apical surfaces. Our results now extend that hypothesis to reveal further roles for cell–cell fusogens in membrane re-organization. Not only do these fusogens remove cell junctions and their associated membranes, but they play more widespread roles in endocytic scission processes for membrane re-distribution.

Our results with AFF-1 suggest that related class II exoplasmic fusogens in other eukaryotes, plants and viruses could also mediate endocytic scission. Furthermore, the possibility should be considered that other structurally-distinct types of cell–cell fusogens, such as the mammalian syncytins or the Myomaker/Myomerger pair[23], could perform similar jobs in endocytic scission and contribute to shaping of large syncytial tissues such as placenta and muscles.

## Methods

**Worm strains, alleles, and transgenes**. All animals used in this study were *Caenorhabditis elegans* hermaphrodites. See Supplementary Table 1 for a complete list of strains used in this study, and Supplementary Table 2 for a list of transgenes. All strains were grown at 20 °C under standard conditions[65] unless otherwise noted. *aff-1* mutants were obtained from homozygous mothers cut open with a razor blade to obtain embryos. Alleles *aff-1(cs232 [aff-1::zf])* and *rdy-2(cs233[rdy-2:: GFP])* were obtained by CRISPR-Cas9[40], using the plasmids pFS149 and pRFR56 respectively as repair templates, and pFS144 and pRFR56 as Cas9 and sgRNA-expressing plasmids. The Self-Excision-Cassette inserted in *aff-1(cs232)* was maintained, since excision resulted in a strong *aff-1* hypomorphic allele by disrupting the *aff-1* 3'UTR. Transgenic animals were generated by injecting N2 with plasmid DNA at 10–30 ng mL⁻¹ together with fluorescent markers and pSK + to a total DNA concentration of 150–200 ng mL⁻¹ (See Supplementary Table 2 for details). *lin-48pro* drives expression in the duct cell beginning at the 2–3-fold stage[66]. *grl-2pro* drives expression in the duct and the pore cell beginning at the 1.5-fold stage[67].

**Plasmids**. The 5.4 kb *aff-1* promoter was amplified by polymerase chain reaction (PCR) from fosmid WRM0615dE03. For CRISPR/Cas9 genome editing, *aff-1*-specific guide RNA: 5′-ttactaaaagctcattcaca-3′ and *rdy-2*-specific guide RNA: 5′-gatcaaacggtgagtgcacg-3′. The repair constructs were both derived from pDD282[40]. For *aff-1::ZF1* genome editing, *GFP* coding sequence was replaced by ZF1 sequence PCR amplified from pJN601[39] with oFS144 and oFS145 and *3xFlag* was removed. Homology arms were PCR amplified by oFS-142/oFS-143 and oFS-148/oFS-149 from the fosmid WRM0615dE03. The self-excision cassette was PCR amplified by oFS-146/oFS-147 and the vector backbone by oFS-150/oFS-151. All PCR fragments were assembled using NEBuilder® HiFi DNA Assembly Master Mix to obtain pFS146. For *rdy-2::GFP* genome editing, repair plasmid was obtained as describe in ref. [40]. Homology arms were obtained by PCR amplification with oFS-167/oFS-168 and oFS-169/oFS-170 from the fosmid WRM0636A_A04, and pDD282 was digested with AvrII and SpeI. All double stranded DNA fragments were assembled using NEBuilder® HiFi DNA Assembly Master Mix. In the resulting plasmid a mutation was inserted in the protospacer adjacent motif (PAM) sequence with NEB Q5® Site-Directed Mutagenesis Kit with oFS-171 and oFS-172 to obtain pRFR56.

All plasmid maps and sequences are available on demand.

See Supplementary Table 3 for a complete list of oligos used in this study and Supplementary Table 4 for a complete list of plasmids.

**Confocal microscopy and image analysis**. Specimens were mounted on a 5% Agar Noble, 20 mM Sodium Azide pad in a drop of 20 mM Levamisole in M9 Buffer. Fluorescent and differential interference contrast images were captured on a compound Zeiss Axioskop fitted with a Leica DFC360 FX camera or with a Leica TCS SP8 confocal microscope. For experiments not involving pixel intensity quantification, confocal laser powers were set to 0.2–5%, and HyD confocal detector sensitivities were set below pixels saturation level in the region of interest (ROI). GFP fused proteins were detected with a 488 nm laser, with a HyD confocal detector set to 490–546 nm. mCherry and mRFP fused proteins were detected with a 552 nm laser and a HyD confocal detector set to 580–670 nm. FM4-64 dye was detected with a 514 nm laser set to 1% power and a HyD confocal detector set to 650–795 nm (Supplementary Fig. 6) or 700–795 nm to limit mCherry bleach through effect (Fig. 6d) or a PMT confocal detector set to 650–795 nm for FRAP experiment (Fig. 5d). For FM4-64 quantification in presence of an mCherry dye, 488 nm laser set to 3% power was used to avoid mCherry bleach through effect (Fig. 5b, c) with a HyD confocal detector set to 700–795 nm. For Fig. 3a, Super-resolution images were obtained with a Leica STED 3 × Super-Resolution Microscope. Images were processed and merged using ImageJ. Auto-fusion was assessed with AJM-1::GFP. Lumen length and apical domain width were assessed with RDY-2::GFP and measured with the Free Hand Line tool in ImageJ by a researcher blinded to genotypes. At least seven animals per genotype were measured and each genotype was treated as an independent sample. Non-parametric statistical tests were used to avoid assumptions about data normality and variance. Auto-fusion and *aff-1* expression data were compared between genotypes by a one-tailed Fisher's Exact test. Lumen measurement distributions were compared by a two-tailed Mann–Whitney *U*-test. All data were analyzed and plotted using Graphpad Prism. AFF-1::mCherry localization analysis was measured with Volocity (Perkim Elmer). The duct cell area was drawn coarsely using the free hand tool, and the three-dimensional duct object was delimited with a threshold of 20–100% pixel intensity. The AFF-1::mCherry objects were counted with the same threshold. The objects exclusively inside the cell volume were subtracted from the objects overlapping the cell volume to estimate the number of objects at the basal surface of the cell. All images and graphics were assembled with Adobe Illustrator CS6.

**Temperature-sensitive allele and heat-shock experiments**. For experiments using *sos-1(cs41ts)* and *dyn-1(ky51ts)*, P0 homozygous hermaphrodites were shifted to 25 °C as young adults, 24–48 h prior to F1 observation. For stage-specific *aff-1::zf1* knock-down experiments, embryos were staged based on morphological criteria and heat-shock was applied for 30 min at 34 °C, followed by one hour recovery at 20 °C, repeated three times. L1 specimens were observed 1–3 h after hatching.

**Serial section transmission electron microscopy**. *aff-1(tm2214)* L1 larvae were prepared by high-pressure freezing and freeze substitution into 2% osmium tetroxide, 0.1% uranyl acetate, and 2% $H_2O$ in acetone[68]. Control *him-5(e1490)* L1 larvae were prepared by high-pressure freezing and freeze substitution into 2% PFA, 2% glutaraldehyde, 4% $H_2O$ in acetone, and postfixed in 2% osmium tetroxide in acetone. Specimens were rinsed and embedded into LX112 resin[69]. Serial thin sections on slot grids were post stained in 2% uranyl acetate. Images were collected on a JEOL-1010 transmission electron microscope, processed in ImageJ and pseudocolored in Adobe Illustrator CS6. Four *aff-1*, two *him-5* and two archival N2 L1 specimens were analyzed. Images of the N2 L1 specimen in Fig. 5a were kindly provided by Nichol Thomson (MRC/LMB) and are publicly available at www.wormimage.org. For excretory duct tube diameter measurement, we used the free hand line tool on ImageJ. Average tube diameter was evaluated on serial sections for each specimen (*n* slices ≥ 6) to calculate a global average diameter for each genotype.

**FM4-64 dye assays**. FM4-64 dye was obtained from Thermo-Fisher scientific (catalog #T-13320) and diluted in M9 buffer to a final concentration of 100 μg mL$^{-1}$. L1 or L4 larvae were soaked in dye solution at 20 °C for the time indicated. Larvae were briefly rinsed in a bath of M9 buffer and transferred to an NGM plate, with OP50, for a 30 min recovery time. Confocal observations were made during the next 30 min after recovery. Dye penetration into the duct and pore cells (Fig. 5b and Supplementary Fig. 6c) was quantified with Volocity (Perkim Elmer). The ROI was drawn coarsely with the free hand tool, and a threshold of 20–100% pixel intensity was applied to define the three-dimensional duct and pore cell bodies within the image stack. The same threshold was used to define FM4-64 objects. The sum of pixel intensities for all the FM4-64 objects overlapping with the cell body object was used to estimate dye entry. Dye penetration into the duct cell (Supplementary Fig. 6a) was quantified using ImageJ and confocal Z-projections. For duct specific measurement, the excretory duct region was selected with the free hand tool, and the total intensity of that area was used to estimate dye entry. Measurements were made on at least five animals per genotype per experiment, wild-type and mutant specimens were analyzed in parallel, and distributions were compared by a non-parametric two-tailed Mann–Whitney *U*-test. All data were analyzed and plotted using Graphpad Prism. For analysis of AFF-1::mCherry localization, 19 worms expressing the transgene *aff-1pro::AFF-1::mCherry* and 16 WT worms were imaged. Data were analyzed in parallel after image name randomization with ImageJ, so that the researcher scoring them was blinded to genotype. The numbers and positions of FM4-64 containing compartments in each image were counted first. Next, the AFF-1::mCherry signal channel was revealed to estimate its position compared to the FM4-64 position. The number of FM4-64 compartments was similar between the two genotypes (3.2 ± 1.3 in *aff-1pro::AFF-1::mCherry* and 3.2 ± 1.2 in WT).

**Fluorescence recovery after photobleaching (FRAP)**. After a 30 min exposure to 100 μg mL$^{-1}$ FM4-64 in M9 buffer, L1 specimens were mounted on 10% agarose pads containing 20 mM sodium azide and 10 mM levamisole in M9. FRAP was performed using Leica Application Suite X software FRAP module on a Leica TCS SP8 MP confocal microscope. A bleach ROI was defined within the wizard, and mean fluorescence intensity within the ROI was measured at specified intervals. The following experimental time-course was used: 20 pre-bleach frames every 0.6 s, 10 bleach frames every 0.6 s, and 90 post-bleach frames every 2.0 s. Pre- and post-bleach laser intensity was set to 1% and bleach laser intensity was set to 100%. To correct for additional bleaching during the post-bleach phase, a double normalization method was applied[70]. Average pre-bleach whole-image intensity, divided by the whole-image intensity at each time point in the post-bleach period, was multiplied to the FRAP ROI intensity at that time point. Before this operation, both whole-image and FRAP ROI data were subtracted by base intensity. FRAP plots were created and analyzed using Graphpad Prism.

**Data availability**. The datasets generated during and/or analyzed during the current study are available from the corresponding author on reasonable request.

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

## Acknowledgements

We thank Ken Nguyen, Leslie Gunther-Cummins, and Geoff Perumal for help with electron microscopy, Benjamin Podbilewicz and Max Heiman for providing reagents,

Benjamin Podbilewicz, Barth Grant, Chris Rocheleau, Sergio Grinstein, Bob Doms, Mickey Marks, and members of the UPenn *C. elegans* community for helpful discussions and advice, Jennifer Cohen for artwork, Rachel Forman-Rubinsky for technical assistance, and Piya Ghose and Shai Shaham for sharing unpublished data. We thank Jonathan Hodgkin for help in transferring the files of Nichol Thomson (MRC/LMB) to the Hall lab, for sharing on www.wormimage.org. Some strains were provided by the Caenorhabiditis Genetics Center (CGC), which is funded by the NIH Office of Research Infrastructure Programs (P40 OD01440). This work was funded by National Institutes of Health grants R01GM58540 to M.V.S. and OD010943 to D.H.H. (with permission).

## Author contributions

F.S. and M.V.S. designed the study and wrote the manuscript. F.S. performed most of the experiments and data analysis. D.H.H. provided all material and facilities for all transmitted electronic microscopy sample preparation, and participated in TEM data analysis and manuscript preparation.

## Additional information

**Competing interests:** The authors declare no competing interests.

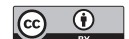

