## [Peer Review File · Nature Communications]

Reviewer #1 (Remarks to the Author):

The manuscript by Soulavie et al. describes novel AFF-1-mediated fission, required for excretory duct elongation during *C. elegans* development. A previously described role for AFF-1 function includes duct cell auto-fusion and seam disappearance. In this research, the authors demonstrate a surprising new role for AFF-1 in excretory duct morphogenesis and in particular elongation. In addition, they demonstrate that AFF-1 expression in the duct cell is positively regulated by EGF-Ras-ERK signaling pathway. Using elegant inducible AFF-1 depletion they demonstrated that AFF-1 mediated tube elongation is independent from, and occurs only after AFF-1 dependent auto-fusion event. AFF-1::mCherry localization correlates with its dual role; during auto-fusion it localizes to apical junctions, while later during seamless tube cell elongation, it localizes mostly to basal membranes. Electron microscopy and live FM4-64 endocytosis experiments demonstrate that *aff-1* worms have shortened duct cell, with accumulated vesicles near the duct lumen and membranous structures continuous with basal membrane. Finally, *rab-5* and *rab-11* cause morphological and elongation defects of the duct cell respectively. Based on these results, the authors suggest that AFF-1 mediates duct cell elongation by membrane scission scenarios; a) AFF-1 mediates scission of basal membrane endosomes and b) AFF-1 facilitates polarized apical exocytosis to elongate seamless tubes. The manuscript has a clear flow and interesting results for cell and developmental biologists. In particular this work uncovers a novel and exciting function for the exoplasmic fusogen AFF-1: endocytic fission and seamless tube elongation. Similar processes occur in diverse eukaryotic cells and it is conceivable that other exoplasmic fusogens such as Syncytins, HAP2, and Myomaker/micropeptides, may have similar endocytic and tube elongation functions in placenta, gametes and muscles respectively. The manuscript can be accepted for publication after some minor rewriting as suggested in the specific comments described below.

Major comments:

1. Line 156 (Fig 4A). It is not clear how AFF-1 may have an "exocytic scission mechanism" this is too speculative and distracting from the main conclusions and proposed mechanism. This effect on exocytosis could be indirect due to a scission failure. If endocytosis (fission) fails then the basolateral membrane will not be able to undergo transfer to the apical domain by transcytosis (Ref 15). AFF-1 could redistribute the membranes from the basolateral membrane to the apical (luminal) domain and a block in endocytosis can inhibit both elongation and expansion of the apical domain due to failure in AFF-1-mediated scission.
2. For simplicity it may be a good idea to remove the proposed "exocytic scission" function of AFF-1 and to focus on AFF-1-mediated endocytic scission followed by transcytosis. In Fig 4A remove the exocytic part (grey) and keep all the lumens (exoplasmic spaces; white)

Minor comments:

3. Line 26: the authors may want to mention one or two specific examples of diseases associated with loss of narrow capillaries.
4. Figure 1(O): for *grl-2pro::AFF-1*, apparently there are 7 and not 17 triangles (worms) for the experiment.
5. Figure 3J-L: please add time post exposure to FM4-64 dye (10 min).
6. The results of TEM and FM4-64 endocytosis, suggest that AFF-1 is required for endocytic scission. For FM4-64 staining, is it possible to measure the staining dynamics in mutant worms with reduced endocytosis and compare it to *aff-1* worms?
7. Line 83. The authors may want to discuss an alternative hypothesis that may explain their observations. Maybe it is trafficking defects due to excess membranes as a result of auto-fusion failure what causes the fission defects. This was previously suggested to explain excess membranes and vesicular organelles observed in *eff-1* mutants observed by TEM (Shemer et al., 2004). Then they can explain that the fact that they can separate the auto-fusion defects from the reduced lumen length and expansion of apical width supports independent functions for AFF-1 (line 90-100).
8. It has been shown that EFF-1 is necessary and sufficient to initiate membrane fusion but also to

expand the fusion pore from microfusion (nanometers) to macrofusion (micrometers). Maybe this expansion involves endocytosis. Indeed, using TEM very small vesicles were identified in wildtype embryonic epidermis during cell-cell fusion in *C. elegans* (Mohler et al., 1998). These vesicles may be the result of endocytosis and fission mediated by EFF-1. Interestingly, similar vesicles were not observed in other fusion events neither in the tail (Nguyen et al., 1999) nor in the pharynx and hypodermis (Shemer et al., 2004).

9. Lines 105-115 may be moved to line 123. The authors may want to add a conclusion sentence for the TEM (Fig 2L-Q) and then follow with the AFF-1-mCherry experiments (Fig 3A-D).

10. Line 127. Are the membrane inclusions inside and discontinuous from the PM? Is this a problem with endocytosis (fission) and/or recycling (exocytosis) or both? Maybe FRAP could determine whether the membrane tubules have pinched off from the PM or are continuous. Also serial sections TEM could differentiate these alternative explanations.

11. Line 135. Use "suggest" instead of "indicate"?

12. Lines 136-138. Again, how can the authors be sure that the extensive membrane compartments are exposed to the environment without FRAP or analyses of serial sections TEM?

13. Have the authors analyzed AFF-1-mCherry localization in rab-5 and rab-11 mutants?

14. Lines 146-148. An alternative explanation for this section is that rab-5 is required for AFF-1 trafficking (as suggested for EFF-1 in embryonic hypodermis; Ref 22) and this affects the diameter of the lumen.

15. Another alternative scenario is that AFF-1 could be present in extracellular vesicles in the lumen of the duct and required for fusion to the apical PM (lumen) to participate in tube elongation (see for example Ref 37).

16. Line 169. Ghose et al., show a function for EFF-1 in phagocytic scission and not endocytosis. The authors may want to revise this.

17. Line 257. Microgram/ml (?)

References

Mohler, W. A., Simske, J. S., Williams-Masson, E. M., Hardin, J. D. and White, J. G. (1998). Dynamics and ultrastructure of developmental cell fusions in the *Caenorhabditis elegans* hypodermis. *Curr. Biol.* 8, 1087-1090.

Nguyen, C. Q., Hall, D. H., Yang, Y. and Fitch, D. H. A. (1999). Morphogenesis of the *Caenorhabditis elegans* male tail tip. *Dev. Biol.* 207, 86-106.

Shemer, G., Suissa, M., Kolotuev, I., Nguyen, K. C. Q., Hall, D. H. and Podbilewicz, B. (2004). EFF-1 is sufficient to initiate and execute tissue-specific cell fusion in *C. elegans*. *Curr Biol* 14, 1587-1591.

Reviewer: Benjamin Podbilewicz

--

Reviewer #2 (Remarks to the Author):

The manuscript submitted by Sundaram and colleagues describes a very interesting set of experiments which leads them to propose the exoplasmic fusion protein AFF-1 is required for endocytic scission and seamless tube elongation in *C. elegans*. This is, to my knowledge, the first time a transmembrane protein involved in exoplasmic fusion events (or any protein with similar function) is proposed to function in endocytic scission!

The authors have used a large number of different experiments to come to this conclusion. They used quantitative analyses in mutant and wild type worms, they used degradation methods to generate a protein loss at a given time point (elegant experiment not done so often since novel), and they used high resolution TEM analysis to look at apical membrane biogenesis and at basolateral luminal pockets. The study appears to be carefully done, and the conclusions are rather

provocative. In no case has a fusion protein been involved in endocytic scission events, and the authors raise the possibility that many exoplasmic fusogens could play broad roles in intracellular membrane merging events.

I think this is a very interesting observation that should be published in Nat Comm. One important experiment would add, however, much credit to the proposed model. Did the authors try to localize AFF-1 to the endocytic vesicle fusion neck? This is a key experiment, because otherwise indirect roles of AFF-1 might provoke the phenotype.

--

Reviewer #3 (Remarks to the Author):

In this paper the authors explore the role of the exoplasmic fusogenic protein AFF-1 in development of the intracellular lumen of a seamless unicellular tube in the *C.elegans* excretory duct. They used elegant genetic/cell biology techniques to show that AFF-1 is necessary for duct auto-fusion and length-wise development. In *aff-1* mutants a blocked endocytic intermediate (basal membrane inclusions) developed. The authors conclude that AFF-1 is involved in endocytic scission at the basal membrane and that basal endocytosis coupled to apically directed exocytosis is required for tube formation.

1. It would be helpful to have a diagram of a whole *C.elegans* to show where the excretory duct actually is.
2. The idea that AFF-1 mediates exoplasmic endocytosis is potentially very exciting. The evidence that AFF-1 mediates exoplasmic endocytosis is: (i) there are membrane inclusions contiguous with the basal plasma membrane in *aff-1* mutants (ii) upon degradation of AFF-1 uptake of FM4-64 is reduced in larval filling assays. So the problem here is that the evidence that AFF-1 is directly mediating exoplasmic endocytosis is circumstantial and rather weak. The authors need at least direct evidence (e.g. immune-EM) that AFF-1 is localized to endocytic membrane invaginations and, ideally, that AFF-1 can mediate scission as they claim.

Response to Reviewers

Thank you very much for the generally enthusiastic comments and useful suggestions for improving our manuscript. We have generated new data and revised the text to address your concerns, as summarized and then detailed point-by-point below (in red). Major changes are highlighted in yellow in the revised manuscript.

New experiments:

1. Confocal imaging data to show that AFF-1::mCherry concentrates near the neck of endocytic membrane invaginations, and therefore is suitably positioned to mediate scission (Fig. 6d,e).
2. FRAP data to show that basal internal membrane inclusions recover rapidly from photobleaching, consistent with being connected to the basal plasma membrane (Figs. 5d and S6).

Other changes to text and figure organization:

The original manuscript had been prepared for another journal and therefore was in a very short format, with only 4 figures.

1. We now have 7 main figures in order to include the new data and arrange the prior figure panels more logically.
2. We have 6 supplemental figures, including Figs. S5 and S6 showing more of our serial section TEM and FRAP data as requested by Reviewer 1.
3. We broke the manuscript up into separate Introduction, Results and Discussion sections, and added additional background and discussion of models to address Reviewer comments.

Reviewer #1

The manuscript can be accepted for publication after some minor rewriting as suggested in the specific comments described below.

Thank you for the supportive comments and many helpful experimental and textual suggestions. We have incorporated most of these suggestions, as detailed below.

Major comments:

1. Line 156 (Fig 4A). It is not clear how AFF-1 may have an “exocytic scission mechanism” this is too speculative and distracting from the main conclusions and proposed mechanism. This effect on exocytosis could be indirect due to a scission failure. If endocytosis (fission) fails then the basolateral membrane will not be able to undergo transfer to the apical domain by transcytosis (Ref 15). AFF-1 could redistribute the membranes from the basolateral membrane to the apical (luminal) domain and a block in endocytosis can inhibit both elongation and expansion of the apical domain due to failure in AFF-1-mediated scission.

We agree that a role of AFF-1 in exocytic scission is just one possible hypothesis to explain the accumulation of apical domain close to the duct lumen in *aff-1* mutants. Such accumulation also could be a secondary defect, as the Reviewer suggests, and so far we do not have data to distinguish between these two possibilities.

We changed the Discussion text to say: “We hypothesize that the exocytic block in *aff-1* mutants is an indirect consequence of the endocytic block – for example, Golgi-derived vesicles may accumulate aberrantly due to absence of appropriate partner vesicles for exocytosis. Alternatively, AFF-1 could play a direct role in some exocytic scission mechanism, but further studies will be needed to address that possibility.”

Also, throughout the text, we now specifically use the term “endocytic scission” rather than “vesicle scission” to describe our results and model.

2. For simplicity it may be a good idea to remove the proposed “exocytic scission” function of AFF-1 and to focus on AFF-1-mediated endocytic scission followed by transcytosis. In Fig 4A remove the exocytic part (grey) and keep all the lumens (exoplasmic spaces; white)

We removed the exocytic scission model from Figure 7 (Figure 4 in the original submission) and made all the exoplasmic spaces white as suggested.

Minor comments:

3. Line 26: the authors may want to mention one or two specific examples of diseases associated with loss of narrow capillaries.

We added some examples in the Introduction: “Loss of narrow capillaries is associated with cardiovascular and neurological syndromes such as small vessel or microvascular disease, hereditary hemorrhagic telangiectasia (HHT) and cerebral cavernous malformation (CCM)”

*4. Figure 1(O): for *grl-2pro::AFF-1*, apparently there are 7 and not 17 triangles (worms) for the experiment.*

Thank you for noticing this error. We fixed the panel, which is now Figure S3 d.

5. Figure 3J-L: please add time post exposure to FM4-64 dye (10 min).

The exposure time has been added (now Figure 5b).

6. The results of TEM and FM4-64 endocytosis, suggest that AFF-1 is required for endocytic scission. For FM4-64 staining, is it possible to measure the staining dynamics in mutant worms with reduced endocytosis and compare it to *aff-1* worms?

Unfortunately, it is not clear what mutants would be appropriate for such an analysis. Our results suggest that endocytosis in the duct depends on a clathrin-independent pathway, but the identity of that pathway is not yet known.

7. Line 83. The authors may want to discuss an alternative hypothesis that may explain their observations. Maybe it is trafficking defects due to excess membranes as a result of auto-fusion failure what causes the fission defects. This was previously suggested to explain excess membranes and vesicular organelles observed in *eff-1* mutants observed by TEM (Shemer et al., 2004). Then they can explain that the fact that they can separate the auto-fusion defects from the reduced lumen length and expansion of apical width supports independent functions for AFF-1 (line 90-100).

We changed the beginning of this section to say: “*aff-1* mutant apical trafficking defects could be a secondary consequence of auto-fusion failure, as previously proposed for *eff-1* mutants (Shemer et al 2004), or could reflect a direct role for AFF-1 in membrane trafficking events. To distinguish between these possibilities, we...”

We also mention the abnormal vesicles in *eff-1* and other mutants in the Discussion: “Several fusogen mutants, including *C. elegans eff-1* and *Tetrahymena hap2*, have previously been found to accumulate abnormal vesicles near unfused plasma membranes, but these vesicles were proposed to be secondary consequences of fusion failure (Cole et al., 2014; Shemer et al., 2004).”

8. It has been shown that EFF-1 is necessary and sufficient to initiate membrane fusion but also to expand the fusion pore from microfusion (nanometers) to macrofusion (micrometers). Maybe this expansion involves endocytosis. Indeed, using TEM very small vesicles were identified in wildtype embryonic epidermis during cell-cell fusion in *C. elegans* (Mohler et al., 1998). These vesicles may be the result of endocytosis and fission mediated by EFF-1. Interestingly, similar vesicles were not observed in other fusion events neither in the tail (Nguyen et al., 1999) nor in the pharynx and hypodermis (Shemer et al., 2004).

We now cite these data in the Discussion: “Cell-cell fusion and endocytic fusion could be mechanistically linked in some cases, since cell-cell fusion removes plasma membrane, some of which may need to be endocytosed and recycled (Podbilewicz 2000); indeed, vesicles have been observed near some (though not

all) fusing plasma membranes in *C. elegans* (Mohler et al 1998; Nguyen et al, 1999; Shemer et al 2004).”

9. Lines 105-115 may be moved to line 123. The authors may want to add a conclusion sentence for the TEM (Fig 2L-Q) and then follow with the AFF-1-mCherry experiments (Fig 3A-D).

We followed the Reviewer’s suggestion to move the description of AFF-1::mCherry to after the description of the TEM results. The original figure 2L-Q is now Figure 3d, and AFF-1::mCherry experiments have been moved to Figure 6.

10. Line 127. Are the membrane inclusions inside and discontinuous from the PM? Is this a problem with endocytosis (fission) and/or recycling (exocytosis) or both? Maybe FRAP could determine whether the membrane tubules have pinched off from the PM or are continuous. Also serial sections TEM could differentiate these alternative explanations.

Both serial section TEM and the FM4-64 experiments support our contention that the tubules are continuous with the PM.

We added a supplemental figure (Figure S5a) to show serial sections of the TEM data. We can trace plasma membrane openings (labeled 1, 2 and 3) across these sections

We also did FRAP analysis on the FM4-64-positive compartments of *aff-1* mutants and showed that these compartments could recover from photo-bleaching (Figures 5d, S6). This experiment argues in favor of our hypothesis.

11. Line 135. Use “suggest” instead of “indicate”?

changed

12. Lines 136-138. Again, how can the authors be sure that the extensive membrane compartments are exposed to the environment without FRAP or analyses of serial sections TEM?

See answer to comment #10

13. Have the authors analyzed AFF-1-mCherry localization in *rab-5* and *rab-11* mutants?

AFF-1::mCherry overexpression appears to have a genetic interaction with these *rab* mutants that has precluded us from doing such experiments; we are trying to obtain a functional CRISPR-generated AFF-1 fusion (or antibody) before pursuing this further.

14. Lines 146-148. An alternative explanation for this section is that rab-5 is required for AFF-1 trafficking (as suggested for EFF-1 in embryonic hypodermis; Ref 22) and this affects the diameter of the lumen.

This is an interesting idea to test in future, once we have an appropriate reagent (see above).

15. Another alternative scenario is that AFF-1 could be present in extracellular vesicles in the lumen of the duct and required for fusion to the apical PM (lumen) to participate in tube elongation (see for example Ref 37).

This is another interesting idea, but we don't favor it since we did not observe vesicles in the duct lumen in any of our electron microscopy observations - neither in *aff-1* mutants nor in *wild type* animals. This model also couldn't explain the basal inclusions that we describe here.

Note that we do see a protrusion of AFF-1::mCherry out of the duct and toward the excretory canal lumen as shown in Figure 6 (and mentioned in the figure legend), so it is possible that duct-derived apical membrane (or vesicles) contribute to the canal cell apical membrane. This is a very interesting observation to follow up on in the future.

16. Line 169. Ghose et al., show a function for EFF-1 in phagocytic scission and not endocytosis. The authors may want to revise this.

We reworded this sentence: "Similarly, Ghose et al (submitted*) have independently shown that the fusogen EFF-1 promotes a specific phagosome sealing event."

17. Line 257. Microgram/ml (?)

The actual value is 100 microgram/ml, corrected in the text.

--

Reviewer #2 (Remarks to the Author):

I think this is a very interesting observation that should be published in Nat Comm. One important experiment would add, however, much credit to the proposed model. Did the authors try to localize AFF-1 to the endocytic vesicle fusion neck? This is a key experiment, because otherwise indirect roles of AFF-1 might provoke the phenotype.

We thank the Reviewer for their supportive comments and this excellent experimental suggestion.

To observe the localization of AFF-1 to the endocytic neck, we used the lipophilic dye FM4-64 in combination with our AFF-1::mCherry transgene. To observe endocytosing vesicles in Wild type, we used a moderate dye exposure time (30') and higher confocal laser power than in the prior experiments. Most (37/59) of the FM4-64-marked vesicles at the basal plasma membrane of the excretory duct were adjacent to a basal spot of AFF-1::mCherry. (Fig. 6d,e). We conclude that AFF-1 is appropriately positioned to mediate endocytic scission.

--

Reviewer #3 (Remarks to the Author):

1. *It would be helpful to have a diagram of a whole C.elegans to show where the excretory duct actually is.*

A diagram of a whole *C. elegans* is now shown in Figure 1b.

2. *The idea that AFF-1 mediates exoplasmic endocytosis is potentially very exciting. The evidence that AFF-1 mediates exoplasmic endocytosis is: (i) there are membrane inclusions contiguous with the basal plasma membrane in aff-1 mutants (ii) upon degradation of AFF-1 uptake of FM4-64 is reduced in larval filling assays. So the problem here is that the evidence that AFF-1 is directly mediating exoplasmic endocytosis is circumstantial and rather weak. The authors need at least direct evidence (e.g. immune-EM) that AFF-1 is localized to endocytic membrane invaginations and, ideally, that AFF-1 can mediate scission as they claim.*

See above response to Reviewer 2. Our new confocal imaging data show that AFF-1 concentrates near the neck of endocytic membrane invaginations, and therefore is suitably positioned to mediate scission. Although we have not conducted biophysical experiments to demonstrate scission activity, there are ample published data showing that AFF-1 and other related fusogens have membrane merging activity. Therefore, we think that it is warranted to publish the extensive data that we have now, and to save further biophysical and mechanistic studies for a future publication.

Reviewer #1 (Remarks to the Author):

The authors have addressed the comments from the reviewers and have added new interesting data on the localization of AFF-1::mCherry and FRAP supporting that AFF-1 is at the right place close to the necks of endocytic invaginations and consistent with the idea that the basal invaginations are continuous with the basolateral plasma membrane. The changes and additions to the text have also improved the introduction, results and discussion of the manuscript. This work is very novel, groundbreaking and will appeal to a broad readership including cell biologists, biomedical scientists and developmental geneticists. I support publication of this revised manuscript “as is”.